# Folic Acid Is Related to Muscle Strength and Vitamin A Is Related to Health-Related Quality of Life: Results of the Korea National Health and Nutrition Examination Survey (KNHANES VII 2016–2018)

**DOI:** 10.3390/nu13103618

**Published:** 2021-10-15

**Authors:** Mee-Ri Lee, Sung Min Jung

**Affiliations:** 1Korea Institute of Health Environment, 87, Seongsui-ro, Seongdong-gu, Seoul 04782, Korea; meerilee83@gmail.com; 2Department of Surgery, Inje University, Ilsan Paik Hospital, 170 Juhwa-ro, IlsanSeo-gu, Goyang-si 10380, Korea

**Keywords:** vitamin A, folic acid, muscle strength, quality of life

## Abstract

This study investigated how folic acid affects muscle strength and the effects of vitamin A on quality of life in adults. Baseline data from the Korea National Health and Nutrition Examination Survey (KNHANES VII 2016–2018) was used to find 6112 adults (ages 19–80) meeting study criteria. The participants were divided into three groups: young adults (ages 19–39), middle-aged (ages 40–64), and elderly (≥65 years). Muscle strength was measured using a digital grip strength dynamometer. The EuroQol five-dimension questionnaire measured quality of life. Associations were assessed using multivariate regression and logistic regression. Vitamins and handgrip strength were divided into low and high groups based on the mean. Although vitamin A and folate levels were related to handgrip strength in all subjects, regression demonstrated a significant association between folate and handgrip strength in the elderly. The odds ratios (OR) of higher handgrip strength were statistically significant for elderly participants with high levels of folate compared to those with low levels (OR: 1.55). Vitamin A was associated with quality of life, especially in the self-care dimension for the elderly. Further longitudinal research is needed to examine the relationship between vitamins and muscle strength, as well as vitamins and quality of life.

## 1. Introduction

Given that the global population over age 60 is expected to double by 2050, there is a growing interest in healthy aging [1]. The aging process increases the risk of physical functional impairment, and improvements in muscle strength and health-related quality of life (HRQOL) are essential factors for healthy aging [2].

Muscle strength has been associated with the risk of all-cause and cardiovascular disease (CVD) mortality [3,4] and is a predictor of new-onset chronic diseases such as hypertension [5] and diabetes [6] in middle-aged and older populations. Handgrip strength (HGS) is a simple anthropometric measurement and an indicator of overall body muscle strength and frailty phenotypes in the elderly [7].

Several observational studies have been conducted to identify whether vitamins contribute to muscle strength and quality of life. Women aged 60 and over with vitamin D deficiency were more likely to have low handgrip strength compared to those with higher vitamin D levels [8]. Vitamin B6 intake was also associated with greater handgrip strength in healthy European older adults with low physical activity [9]. Additionally, higher vitamin E levels are related to handgrip strength in Korean young adults (≤40 years) but not in older adults (40 years) [10]. However, other studies using Korean adults showed an association between vitamin E and handgrip strength in those aged 50 years and older [11]. Small group studies have shown that folic acid affects handgrip strength [12,13,14].

Some randomized controlled trials on the relationship between vitamins and handgrip strength or quality of life were inconsistent, and all interventions used not only single nutrition but also multiple nutrition. Combined supplementation of whey protein, and vitamins D and E improves handgrip strength and quality of life in older adults [15]. However, some randomized trials have shown that long-term antioxidant supplementation (combined vitamin C, E, beta-carotene, selenium, and zinc) has no effect on HRQOL [16]. Vitamin B12 and folic acid supplementation were administered for 2 years, but there was no effect in limiting reduced handgrip strength in Dutch individuals over 65 years of age (*n* = 2919) [17].

Oxidative stress and inflammation have an important role in the age-related musculoskeletal change [18]. So, vitamin A, antioxidant and anti-inflammatory [19] could affect muscle strength. Folic acid intake could have a beneficial effect on muscle by reducing homocysteine [20]. However, there are few studies on vitamin A and fo-late in relation to muscle strength and HRQOL in general population, and most of these studies were conducted with a small number of subjects.

The study of handgrip strength and quality of life among the general population can be especially important, as it allows us to understand how vitamins work differently in young, middle-aged, and older adults. Thus, the objective of this study was to relate vitamin A and folate levels with handgrip strength and HRQOL in general population-based data using the Korean National Health and Nutrition Examination Survey (KNHANES VII 2016–2018), a representative cross-sectional study in Korea.

## 2. Materials and Methods

KNHANES was a nationwide representative survey conducted by the Ministry of Health and Welfare. Data were collected by randomly selecting participants using a complex sample design, in which sampling weights were provided to represent the entire Korean population. Of a total of 19,389 adults aged 19 years and older were, 13,227 were excluded due to lack of data on vitamin A, folate, and handgrip strength; ultimately, 6112 adults were analyzed (Figure 1). Participants were divided into three age groups: young adults (ages 19–39), middle-aged (ages 40–64) and elderly (ages 65–80).

### 2.1. Vitamin A and Folate Measurement

Vitamin A was detected in samples by Agilent1200 (Agilent Technologies Inc., Santa Clara, CA, USA) tools and Chromsystems (Chromsystems Instruments Chemicals GmbH, Gräfelfing, Germany) reagent through high-performance liquid chromatography with flame ionization detector methods.

Folate was detected in samples using ARCHITECT i4000Sr (Abbott Laboratories, Lake Bluff, IL, USA) and ARCHITECT Folate-only reagent with chemiluminescent microparticle immunoassay methods.

### 2.2. Health-Related Quality of Life

HRQOL was assessed using the EuroQol 5-dimension (EQ-5D) instrument, which is composed of five multiple-choice questions and the EQ-5D index. HRQOL was assessed in five areas: mobility, self-care, usual activities, pain/discomfort, and anxiety/depression.

For each question, the participants were asked to choose one of three levels: 1 = “No problem at all”, 2 = “There are slight problems”, or 3 = “There are a lot of problems”. The responses to the five questions were subjected to the score conversion system, using the formula for the EQ-5D index score was provided in a previous study [21] as well as on the KNHANES website (https://knhanes.kdca.go.kr/knhanes/main.do, accessed date (14 October 2021)).

In this study, the EQ-5D index score was converted to a perfect score of 100 points. To analyze the five dimensions of the EQ-5D questionnaire, the response “no problem at all” was changed to “no problem status”, while “some problems” or “many problems” were changed to “problem status”.

### 2.3. Handgrip Strength

Handgrip strength was measured using a digital grip strength dynamometer. Participant grip strength was measured three times on each hand with a rest period of 1 min between each trial. Detailed measurement methods can be found in our previous paper [22]. The handgrip strength of the participants’ dominant hand was defined using the question “Which hand do you mainly use”, and the average of all three handgrip strength measurements was calculated [23].

### 2.4. Covariates

Covariates were obtained using self-reported questionnaires. Education levels were divided into three categories: ≤9, 10–12, and 12 years of schooling. Smoking status was divided into lifelong non-smoker, former smoker, and current smoker. Current smokers were defined as those who had smoked at some point in their lifetime and were currently smoking. Past smokers were defined as those with a prior history of smoking. Alcohol consumption was defined as those who drank at least one drink per month during the past year. Physically active was defined as being active at least 150 min per week at moderate intensity or 75 min per week at a high intensity or some combination of moderate-intensity and high-intensity physical activity. Body mass index (BMI) was calculated as weight divided by height squared (kg/m^2^). Obesity was defined as a BMI of 25 or higher.

### 2.5. Statistical Analysis

We combined raw data from KNHANES VII (2016–2018) using complex sample analysis according to the statistical guidelines provided by the Centers for Disease Control and Prevention.

Differences in sociodemographic and clinical characteristics between each age group were compared using one-way ANOVA or Chi-square test, as appropriate.

The association between vitamin A, folate, and handgrip strength was analyzed in two models using complex samples multivariate regression. Model 1 was adjusted for age and sex. Model 2 was adjusted for Model 1, plus education, alcohol consumption, smoking status, physical activity, and obesity. Vitamin A and folate levels were divided into two groups (≥mean, mean).

Handgrip strength was divided into two groups (≥mean and mean) for total participants, age-based groups, and based on sex. Odds ratios (OR) are calculated by dividing the odds of the occurrence of above-average handgrip strength (in those with above-average vitamin A and folic acid levels) by the odds of the occurrence of above-average handgrip strength (in those with lower-than-average vitamin A and folic acid levels).

The EQ-5D was divided based on the 20th percentile. The association between vitamin A and folate and the binary variable of EQ-5D or five dimensions of EQ-5D was analyzed using complex samples logistic regression analysis.

A *p*-value of 0.05 or less is defined as statistically significant. All statistical analyses and graphs were performed using Stata version 17 (Stata Corp., College Station, TX, USA) and R software, version 4.0.2 (The Comprehensive R Archive Network: http://cran.r-project.org, accessed date (14 October 2021)).

## 3. Results

A total of 6112 participants (19–39 years: 1954, 40–64 years: 2962, 65–80 years: 1196) were included in the analysis. The mean ages of the three groups were 30.4 years, 51.9 years, and 72.2 years, respectively. The mean levels of vitamin A and folate were 0.52 mg/L and 7.37 ng/mL (total participants), 0.47 mg/L and 6.47 ng/mL (young adults), 0.54 mg/L and 7.77 ng/mL (middle-aged), and 0.53 mg/L and 7.88 ng/mL (elderly). The mean hand grip strengths for male and female were 37.55 kg and 22.00 kg in total participants; 40.16 kg and 23.43 kg in young adults; 39.78 kg and 22.62 kg in the middle-aged; 30.72 kg and 17.86 kg in the elderly. The 20th percentile of EQ-5D was 91.3 in the total participants, young adult, and middle-aged groups; it was 77.4 in the elderly group.

The differences in demographic and anthropometric characteristics between the young adult, middle-aged, and elderly groups were compared using chi-square tests and one-way ANOVA, as shown in Table 1. There were statistically significant differences in age, sex, education, smoking status, alcohol consumption, physical activity, vitamin A, folic acid, and EQ-5D in the young adult, middle-aged, and elderly groups (Table 1).

After adjusting for covariates, vitamin A and folate levels were associated with handgrip strength in total participants. Folate was significantly related to handgrip strength in the elderly (Table 2).

After adjustment for covariates, among individuals who had high levels of folate, the OR of having higher handgrip strength was 1.18 (95% confidence interval (CI): 1.03–1.36) compared to those with low levels in all participants (Figure 2). In the young adult and middle-aged groups, there was no association between folate levels and handgrip strength, while higher folate was associated with higher handgrip strength in the elderly (OR (95% CI): 1.55(1.10–2.19)) (Figure 2).

The association between vitamin A, folate, and the EQ−5D index using logistic regression models is presented in Figure 3. In the elderly group, high vitamin A levels were significantly associated with high EQ-5D (OR (95% CI): 1.55 (1.03–2.33)).

The ORs for the association between vitamin A and folate levels and the five dimensions of the EQ-5D according to the multivariate logistic regression analysis are presented in Table 3. In the elderly, higher vitamin A level was the only significant association with no problem of self-care (OR (95% CI): 1.87 (1.09–3.17)), while no significant association was observed in the young adult and middle-aged groups.

## 4. Discussion

This study aimed to assess whether blood levels of vitamin A and folate are associated with handgrip strength and HRQOL in Korean adults. We found that, in the elderly higher blood levels of folate were associated with improved handgrip strength and that higher blood levels of vitamin A were associated with higher EQ-5D index. There was no significant relationship between folate and handgrip strength or Vitamin A and HRQOL in the young adult or middle-aged groups. Similar to our findings, the role of folate in improving handgrip strength has been reported three times previously [12,13,14]. Ao et al. [12] showed that a higher serum folate concentration positively contributed to handgrip strength and lower limb muscle strength; higher plasma homocysteine concentration related to lower handgrip strength in 65 elderly Japanese women. Yeung et al. [14] reported that higher folate intake was associated with higher muscle strength; however, after Bonferroni correction, statistical association was lost in 58 geriatric outpatients living in the Netherlands. Wee [13] demonstrated that blood levels of folic acid were significantly correlated with handgrip and leg strength in diabetes mellitus patients 65 years old in Singapore (*n* = 56).

The relationship between folic acid and handgrip strength may be explained by several biological mechanisms. Blood levels of homocysteine (Hcy) increase with age [24] and elevated Hcy levels are associated with a decline in muscle strength [25,26]. Dietary supplementation with folic acid can reduce Hcy levels [27]; therefore, it would be possible to prevent muscle weakness. In addition, folic acid deficiency has been linked to poor diet [19] and healthier dietary patterns seem to protect against weaker muscle strength in the elderly [28].

In our study, vitamin A is related to handgrip strength in the total population but there was no association when analyzed by age group (three groups). Similar to our results, vitamin A levels have not been associated with a loss of muscle mass in the elderly with type 2 diabetes mellitus [29]. A relationship between vitamin A and muscle strength has not been established. Oxidative stress and inflammation are major contributors to muscle atrophy by modulating protein synthesis and proteolysis [30]. Vitamin A contains powerful antioxidants which provide protection against oxidative stress and inflammation [31]; therefore, this could affect muscle strength. Further large-scale studies are needed in this area. There is a small body of epidemiologic studies suggesting a relationship between vitamin A and HRQOL. All-trans retinol (Vitamin A) is related to self-care, usual activities, and pain/discomfort dimensions in non-institutionalized nonagenarians in Spain (*n* = 20) [32]. Increased serum levels of vitamin A and vitamin D3 (25OHD3) reflect good quality of life in Chinese children with stable asthma (*n* = 117 cases and 129 controls) [33].

The biological mechanism by which vitamin A affects quality of life remains unclear. Vitamin A has been known to play an important role in preventing the aging process [19]. Retinoic acid, a common form of active vitamin A, may be linked to the development of regulatory T cells and enhanced responses to cancer, infection [34,35] and autoimmune diseases such as multiple sclerosis [36], all of which are associated with aging and quality of life.

The present study has important strengths. It is the first study to examine the association between vitamin A and folate levels, handgrip strength, and quality of life using complex sample analysis to represent the entire Korean population. Additionally, we were able to stratify the analysis by age and analyzed the results in three groups (young adults, middle-aged adults, and the elderly).

On the other hand, the data of the present study does have several limitations. First, our data consisted of only Korean participants, and it may be difficult to extend the results to other populations. Second, a reverse relationship was not excluded due to cross-sectional study design.

## 5. Conclusions

We found that, in the elderly higher folate levels may improve handgrip strength, and higher vitamin A levels reflect good HRQOL. Future research for the appropriate management of adults with lower vitamin A and folate levels will help determine the clinical implications of these findings for healthy aging.

## Figures and Tables

**Figure 1 nutrients-13-03618-f001:**
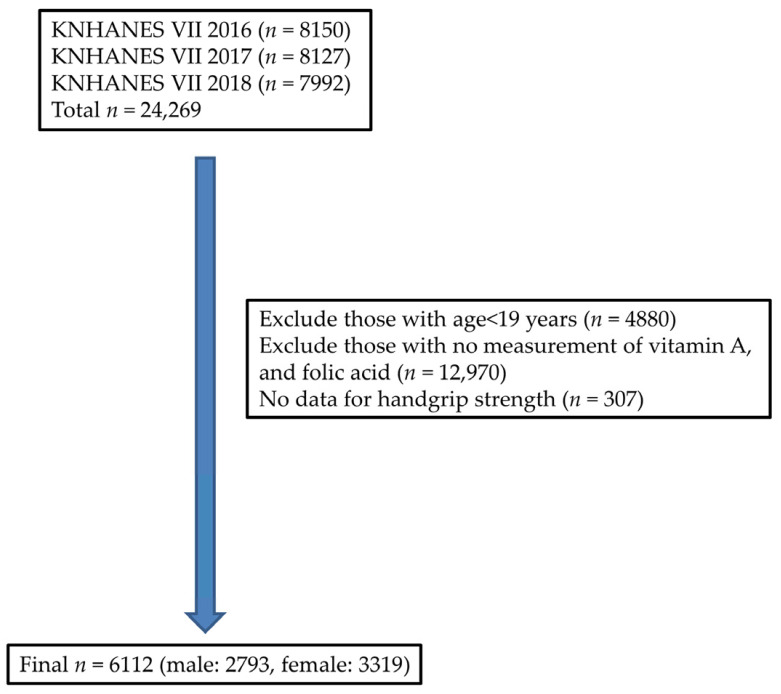
Flow chart shows the number of participants excluded and analyzed.

**Figure 2 nutrients-13-03618-f002:**
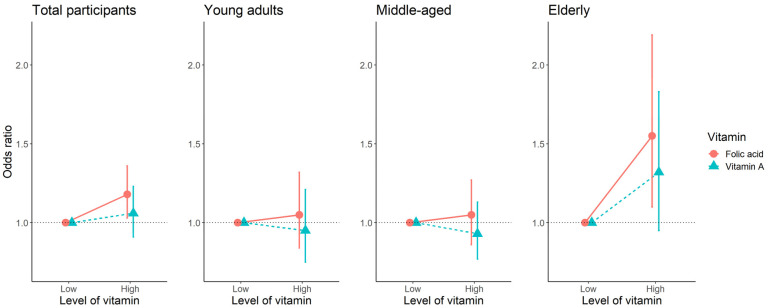
The association between binary variables of vitamin A and folic acid, and handgrip strength using logistic regression adjusted for covariates. Vitamin A: low (0.52 mg/L), high (≥0.52 mg/L) in total participants; low (0.47 mg/L), high (≥0.47 mg/L) in young adults; low (0.54 mg/L), high (≥0.54 mg/L) in middle-aged; low (0.53 mg/L), high (≥0.53 mg/L) in the elderly. Folic acid: low (7.37 ng/mL), high (≥7.37 ng/mL) in total participants; low (6.47 ng/mL), high (≥6.47 ng/mL) in young adults; low (7.77 ng/mL), high (≥7.77 ng/mL) in middle-aged; low (7.88 ng/mL), high (≥7.88 ng/mL) in the elderly. Higher handgrip strength: male ≥ 37.55 kg and female ≥ 22 kg in total participants; male ≥ 40.16 kg and female ≥ 23.43 kg in young adults; male ≥ 39.78 kg and female ≥ 22.62 kg in middle-aged; male ≥ 30.72 kg and female ≥ 17.86 kg in the elderly.

**Figure 3 nutrients-13-03618-f003:**
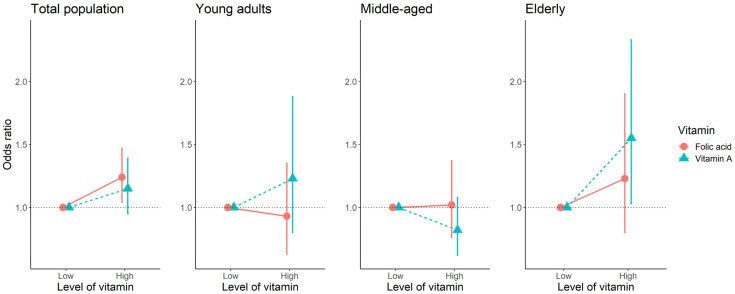
The association between binary variables of vitamin A and folic acid, and EQ-5D using logistic regression. Abbreviations: EQ-5D, EuroQol Five-Dimension Questionnaire; OR, odds ratio; Higher EQ-5D: ≥91.3 in total participant, young adult and middle-aged groups; ≥77.4 in elderly.

**Table 1 nutrients-13-03618-t001:** Participants’ sociodemographic and clinical characteristics.

Variable	Total	19–39	40–64	65+	*p* Value
*n*	6112	1954	2962	1196	
Age, y, mean ±SD	49.1 ± 16.2	30.4 ± 6.1	51.9 ± 7.0	72.5 ± 5.1	**0.001**
Sex, *n* (%) female	3319(49.7)	1053(47.2)	1651(50.6)	615(52.7)	**0.004**
Education, *n* (%)					
Low	1388(19.6)	22(0.8)	601(18.4)	765(64.8)	**0.001**
Medium	1626(26.9)	317(15.8)	1080(37.8)	229(19.9)	
High	2853(53.5)	1550(83.4)	1159(43.8)	144(15.3)	
Not response	245	65	122	58	
Smoking status, *n* (%)					
Never smoker	3573(57.5)	1166(58.7)	1728(55.7)	679(59.8)	**0.001**
Former smoker	1292(22.0)	308(17.3)	606(22.7)	378(30.7)	
Current smoker	1189(20.5)	463(24.0)	608(21.6)	118(9.5)	
No response	58	37	20	21	
Alcohol consumption, *n* (%)					
Non-drinker	2646(40.7)	634(31.1)	1263(40.3)	749(63.5)	**0.001**
Alcohol drinker	3420(59.3)	1310(68.9)	1680(59.7)	430(36.5)	
No response	46	10	19	17	
Physical activity, *n* (%)					
No	3187(52.0)	855(43.1)	1583(54.1)	749(66.0)	**0.001**
Yes	2678(48.0)	1033(56.9)	1262(45.9)	383(34.0)	
No response	247	66	117	64	
Dominant handgrip, kg	29.10 ± 10.21	31.15 ± 10.35	29.77 ± 9.90	24.11 ± 9.08	**0.001**
BMI, *n* (%)					
25	4011(65.6)	1366(69.3)	1899(63.6)	746(62.9)	0.125
≥25	2092(34.4)	586(30.7)	1061(36.4)	445(37.1)	
No response	9	2	2	5	
Vitamin A, mg/L	0.52 ± 0.19	0.47 ± 0.17	0.54 ± 0.19	0.53 ± 0.20	**0.001**
Folic acid, ng/mL	7.37 ± 3.57	6.47 ± 3.40	7.77 ± 3.45	7.88 ± 3.87	**0.001**
EQ-5D	95.27 ± 10.06	97.65 ± 5.54	96.46 ± 7.74	88.36 ± 16.12	**0.001**

χ^2^ test and one-way ANOVA were used for categorical and continuous variables, respectively. Bold numbers highlight the statistical significance. Abbreviations: BMI, body mass index; EQ-5D, EuroQol 5-dimension; SD, standard deviation.

**Table 2 nutrients-13-03618-t002:** Models of association between level of vitamin A and folic acid and handgrip strength.

		Handgrip Strength	
		β	*p* Value
Total (*n* = 6112)	Vitamin A		
	Model 1	0.91 ± 0.22	**0.001**
	Model 2	0.52 ± 0.22	**0.019**
	Folic acid		
	Model 1	0.62 ± 0.21	**0.004**
	Model 2	0.74 ± 0.21	**0.001**
19–39 (*n* = 1954)	Vitamin A		
	Model 1	0.25 ± 0.35	0.475
	Model 2	−0.19 ± 0.36	0.605
	Folic acid		
	Model 1	−0.18 ± 0.33	0.584
	Model 2	0.12 ± 0.33	0.724
40–64 (*n* = 2962)	Vitamin A		
	Model 1	−0.20 ± 0.27	0.462
	Model 2	−0.37 ± 0.28	0.181
	Folic acid		
	Model 1	0.01 ± 0.28	0.969
	Model 2	0.04 ± 0.28	0.887
65+ (*n* = 1196)	Vitamin A		
	Model 1	0.72 ± 0.41	0.083
	Model 2	0.75 ± 0.41	0.065
	Folic acid		
	Model 1	1.09 ± 0.40	**0.006**
	Model 2	0.91 ± 0.43	**0.035**

Bold numbers highlight the statistical significance. Statistical models are as follows: Model 1: adjusted for age and sex; Model 2: Model 1, plus education, alcohol consumption, smoking status, physical activity, and obesity.

**Table 3 nutrients-13-03618-t003:** Associations between vitamin A and folic acid and each of the five EQ-5D dimensions using multiple logistic regression analysis.

		Mobility		Self-Care		Usual Activities		Pain/Discomfort		Anxiety/Depression	
		OR(95%CI)		OR(95%CI)		OR(95%CI)		OR(95%CI)		OR(95%CI)	
		No Problem	Problems (Ref.)	No Problem	Problems (Ref.)	No Problem	Problems (Ref.)	No Problem	Problems (Ref.)	No Problem	Problems (Ref.)
Total	Vitamin A	1.11(0.88–1.40)	1	1.47(0.99–2.20)	1	**1.36(1.01–1.84)**	1	0.96(0.82–1.13)	1	1.12(0.89–1.41)	1
	Folic acid	**1.27(1.02–1.58)**	1	**1.65(1.08–2.50)**	1	1.22(0.91–1.63)	1	0.98(0.82–1.16)	1	1.18(0.93–1.49)	1
19–39	Vitamin A	1.95(0.82–4.60)	1	0.20(0.01–7.24)	1	1.05(0.34–3.25)	1	1.20(0.81–1.80)	1	1.24(0.78–1.97)	1
	Folic acid	0.91(0.46–1.80)	1	3.66(0.23–57.47)	1	0.68(0.27–1.74)	1	0.79(0.57–1.08)	1	1.01(0.65–1.57)	1
40–64	Vitamin A	0.71(0.50–1.03)	1	1.04(0.50–2.13)	1	1.06(0.61–1.84)	1	**0.75(0.59–0.94)**	1	0.89(0.63–1.26)	1
	Folic acid	1.17(0.81–1.69)	1	1.50(0.73–3.11)	1	1.17(0.70–1.96)	1	0.90(0.68–1.18)	1	0.89(0.61–1.30)	1
65+	Vitamin A	1.22(0.87–1.72)	1	**1.87(1.09–3.17)**	1	1.31(0.87–1.96)	1	1.34(0.97–1.84)	1	1.29(0.84–1.98)	1
	Folic acid	1.32(0.95–1.84)	1	1.54(0.88–2.72)	1	0.98(0.65–1.47)	1	0.99(0.70–1.40)	1	1.29(0.84–1.99)	1

Bold numbers highlight the statistical significance. Abbreviations: CI, confidence interval; EQ-5D, EuroQol five-dimension questionnaire; OR, odds ratio; Ref, reference.

## Data Availability

The data from the KNHANES are available on request by email or by visiting the Korea National Health and Nutrition Examination Survey website (https://knhanes.kdca.go.kr/knhanes/main.do, accessed date (14 October 2021)). These data are free of charge for academic research.

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
