# Peer review of "Folic Acid Is Related to Muscle Strength and Vitamin A Is Related to Health-Related Quality of Life: Results of the Korea National Health and Nutrition Examination Survey (KNHANES VII 2016–2018)"

_nutrients, 2021, doi:10.3390/nu13103618_

Round 1

Reviewer 1 Report

 The manuscript presents an interesting topic. Generally, the text is well written. However, I have some issues:

  1. The title shows that there are relations between (both) folic acid and vitamin A and muscle strength and quality of life. It is not exactly what there is in the conclusion.
  2. Why Authors divided participants into 2 groups? why young are in the same group as middle-aged? Analyzed parameters are different in 19 years men/women than in 64 years men/women. Groups differ significantly in numbers (19-64 n=4916; 65+ n=1196); in my opinion, three groups should be analyzed (young, middle-aged, elderly);
  3. Authors should show (on an example) how the odds ratio was calculated (fig. 2)?
  4. Conclusion: the first sentence should be removed because it is not a conclusion.
  5. Abstract: the first sentence „This study investigated the effect of vitamin A and folate on…” should be changed because the Authors analyzed relations not the effect.

Reviewer 2 Report

The authors should address the possible relationship between vitamin a and acid folic with muscle strength.

section 2.4
which questionnaire was used to determine the dominant hand?
which of the three attempts was used as data? all three or the best of the three?

Line- 180- Correct ".."

Section 4. Discussion

The manuscript is well done and with a fine analysis, however, the discussion should go accordingly. The relationship between Handgrip strength and the vitamins analysed should be further justified. Folic acid is known to be related. However, more depth is needed to justify the relationship of vitamin A to strength.

Round 2

Reviewer 1 Report

In my opinion, the Authors improved the text and in this form, it can be accepted.